# Host proteostasis modulates influenza evolution

Angela M Phillips[1], Luna O Gonzalez[2], Emmanuel E Nekongo[1],
Anna I Ponomarenko[1], Sean M McHugh[3], Vincent L Butty[4], Stuart S Levine[4],
Yu-Shan Lin[3], Leonid A Mirny[5,6], Matthew D Shoulders[1]*

[1]Department of Chemistry, Massachusetts Institute of Technology, Cambridge, United States; [2]Department of Mathematics, Massachusetts Institute of Technology, Cambridge, United States; [3]Department of Chemistry, Tufts University, Medford, United States; [4]BioMicro Center, Massachusetts Institute of Technology, Cambridge, United States; [5]Department of Physics, Massachusetts Institute of Technology, Cambridge, United States; [6]Institute for Medical Engineering and Science, Massachusetts Institute of Technology, Cambridge, United States

**Abstract** Predicting and constraining RNA virus evolution require understanding the molecular factors that define the mutational landscape accessible to these pathogens. RNA viruses typically have high mutation rates, resulting in frequent production of protein variants with compromised biophysical properties. Their evolution is necessarily constrained by the consequent challenge to protein folding and function. We hypothesized that host proteostasis mechanisms may be significant determinants of the fitness of viral protein variants, serving as a critical force shaping viral evolution. Here, we test that hypothesis by propagating influenza in host cells displaying chemically-controlled, divergent proteostasis environments. We find that both the nature of selection on the influenza genome and the accessibility of specific mutational trajectories are significantly impacted by host proteostasis. These findings provide new insights into features of host–pathogen interactions that shape viral evolution, and into the potential design of host proteostasis-targeted antiviral therapeutics that are refractory to resistance.

DOI: https://doi.org/10.7554/eLife.28652.001

*For correspondence:
mshoulde@mit.edu

Competing interests: The authors declare that no competing interests exist.

## Introduction

Minimalist pathogens like RNA viruses survive dynamic host environments by virtue of their extreme adaptability. This adaptability is driven by a high rate of genetic variation, mediated by error-prone genome replication (*Sanjuán et al., 2010*). Most missense mutations have deleterious consequences for protein function, often owing to either thermodynamic (reduced stability of the native state or enhanced stability of unfolded/misfolded states) or kinetic (slow folding or enhanced misfolding/ aggregation) effects on folding (*DePristo et al., 2005*). In the context of viruses, these phenomena may underpin the observation that the distribution of mutational fitness effects can be largely accounted for by considering protein folding biophysics (*Wylie and Shakhnovich, 2011*; *Chéron et al., 2016*; *Tokuriki and Tawfik, 2009a*). Indeed, stable proteins tend to be more evolvable, as any given missense mutation is less likely to severely disrupt protein folding or structure (*Bloom et al., 2006*; *Gong et al., 2013*).

In cells, protein folding challenges are addressed by proteostasis networks composed of chaperones and quality control factors that work in concert to shepherd nascent proteins to folded, functional conformations (*Balch et al., 2008*; *Hartl et al., 2011*; *Powers and Balch, 2013*). Important work focused primarily on the Hsp90 chaperone has suggested a critical role for chaperones in modulating the evolution of their endogenous clients, (*Cowen and Lindquist, 2005*; *Queitsch et al.,*

**eLife digest** Influenza viruses, commonly called flu, can evade our immune system and develop resistance to treatments by changing frequently. Specifically, mutations in their genome cause influenza proteins to change in ways that can help the virus evade our defences. However, these mutations come at a cost and can prevent the viral proteins from forming functional and stable three-dimensional shapes – a process known as protein folding – thereby hampering the virus' ability to replicate.

In human cells, proteins called chaperones can help our other proteins fold properly. Influenza viruses do not have their own chaperones and, instead, hijack those of their host. Host chaperones are therefore crucial to the virus' ability to replicate. However, until now, it was not known if host chaperones can influence how these viruses evolve.

Here, Phillips et al. used mammalian cells to study how host chaperones affect an evolving influenza population. First, cells were engineered to either have normal chaperone levels, elevated chaperone levels, or inactive chaperones. Next, the H3N2 influenza strain was grown in these different conditions for nearly 200 generations and sequenced to determine how the virus evolved in each distinctive host chaperone environment.

Phillips et al. discovered that host chaperones affect the rate at which mutations accumulate in the influenza population, and also the types of mutations in the influenza genome. For instance, when a chaperone called Hsp90 was inactivated, mutations became prevalent in the viral population more slowly than in cells with normal or elevated chaperone levels. Moreover, some specific mutations fared better in cells with high chaperone levels, whilst others worked better in cells with inactivated chaperones.

These results suggest that influenza evolution is affected by host chaperone levels in complex and important ways. Moreover, whether chaperones will promote or hinder the effects of any single mutation is difficult to predict ahead of time. This discovery is significant, as the chaperones available to influenza can vary in different tissues, organisms and infectious conditions, and may therefore influence the virus' ability to change and evolve in a context-specific manner.

The findings are likely to extend to other viruses such as HIV and Ebola, which also hijack host chaperones for the same purpose. More work is now needed to systematically quantify these effects so that we can better predict how specific chaperones will affect the ability of viruses to adapt, especially in pathologically relevant conditions like fever or viral host-switching. In the future, such insights could help shape the design of treatments to which viruses do not evolve resistance.

DOI: https://doi.org/10.7554/eLife.28652.002

*2002*; *Lachowiec et al., 2015*; *Sangster et al., 2007*, *2008a*, *2008b*; *Rohner et al., 2013*; *Whitesell et al., 2014*; *Geiler-Samerotte et al., 2016*; *Rutherford and Lindquist, 1998*) in part by buffering deleterious effects of non-synonymous mutations. The consequences of Hsp90 activity for protein evolution may be due to Hsp90 directly engaging an evolving client protein (termed a primary effect). Alternatively, the effects of Hsp90 may be secondary, mediated indirectly by Hsp90 influencing the folding of other endogenous clients that themselves engage relevant evolving proteins. For instance, Hsp90-dependent azole resistance in *Candida albicans* is mediated by secondary effects of Hsp90 on calcineurin, an Hsp90 client that regulates responses to environmental stimuli (*Cowen and Lindquist, 2005*). Efforts to look beyond Hsp90 to understand how other components of the metazoan proteostasis machinery modulate evolution (e.g., Hsp40/70 chaperones or protein misfolding stress responses like the heat shock response) have been slowed by the paucity of chemical biology tools to perturb the activities of these systems. However, Tawfik and coworkers have shown that the GroEL/ES chaperonin system can govern the fitness of certain client protein variants in bacteria (*Tokuriki and Tawfik, 2009b*), and computational modeling suggests that other chaperones may also have roles in evolution (*Bogumil and Dagan, 2012*; *Cetinbaş and Shakhnovich, 2013*).

Chaperones and other proteostasis mechanisms are theoretically well-positioned to address the biophysical challenges created by high mutation rates in viruses. Intriguingly, most RNA viruses lack autonomous chaperones or other co-factors to assist their proteins with folding. Instead, viral

proteins engage host chaperones, (*Melville et al., 1999*; *Momose et al., 2002*; *Naito et al., 2007*; *York et al., 2014*; *Watanabe et al., 2010*) and host chaperone inhibitors have been shown to limit the viability of certain RNA viruses (*Geller et al., 2007*; *Heaton et al., 2016*; *Taguwa et al., 2015*; *Chase et al., 2008*; *Geller et al., 2012*). However, the possibility that host chaperones can shape the evolution of viral pathogens has not been investigated.

In summary, it is clear that: (1) high genetic variability is essential to support RNA virus adaptability; (2) missense mutations important for viral adaptation are often biophysically deleterious, constraining the accessible mutational landscape; and (3) many viruses engage host chaperones to fold their proteins. An important but still untested hypothesis is that host proteostasis modulates RNA virus evolutionary trajectories. Experimentally testing this hypothesis requires methods to regulate the host cell's proteostasis network without significantly perturbing cell health or the ability of an RNA virus to propagate. Here, we achieve this goal in the context of long-term influenza propagation by using small molecules to either modulate the heat shock response in a stress-independent manner (*Shoulders et al., 2013*; *Moore et al., 2016*) or to inhibit Hsp90 at sub-lethal concentrations (*Ying et al., 2012*). We find that the resulting perturbations to host proteostasis mechanisms significantly impact both the extent and the nature of selection pressure on the influenza genome. We conclude that host proteostasis is a critical, under-appreciated player in influenza evolution, with significant implications for our ability to predict and prevent the evolution of influenza and other RNA viral pathogens.

## Results

### Small molecule-based strategies create three distinctive host proteostasis environments for influenza evolution experiments

Eukaryotic cells dynamically match proteostasis network capacity to demand via compartment-specific stress responses. Thus, one biologically relevant strategy to create an altered proteostasis environment is to induce such responses. We focused on the heat shock response, (*Åkerfelt et al., 2010*) because numerous influenza proteins must fold and/or function in the cytosol and nucleus (*Watanabe et al., 2010*). Typical methods to induce heat shock factor 1 (HSF1), the master regulator of the heat shock response and thus of cytosolic and nuclear chaperone and quality control protein levels, involve treatment with toxins or acute heat stress. These methods are not useful for our studies because they engender massive protein misfolding stress in host cells that rapidly become apoptotic, preventing influenza propagation. An alternative strategy is to over-express a constitutively active form of HSF1 lacking amino acids 186–202 (termed cHSF1), (*Voellmy, 2005*) but high levels of cHSF1 over-expression outside the physiologically relevant regime are typically toxic (*Ryno et al., 2014*).

Instead, we took a chemical genetic approach, fusing cHSF1 to a destabilized variant of *E. coli* dihydrofolate reductase (DHFR) (*Shoulders et al., 2013*; *Iwamoto et al., 2010*). The DHFR.cHSF1 fusion is targeted for rapid proteasomal degradation and is therefore non-functional (*Figure 1A*), unless cells are treated with the DHFR-stabilizing pharmacologic chaperone trimethoprim (TMP). We created a stable, clonal Madin Darby canine kidney (MDCK) cell line expressing DHFR.cHSF1. In these cells, termed MDCK$^{HSF1}$ cells, we can dosably induce cHSF1 transcriptional activity by TMP treatment in a stress-independent manner within the biologically relevant regime (*Figure 1B*), avoiding cytotoxicity that would be induced by stressors or cHSF1 overexpression and yet still providing robust access to cells expressing enhanced levels of HSF1 targets (*Figure 1—figure supplement 1A–C*). To control for any possible unintended consequences of TMP treatment or expression of a DHFR-fusion protein in our evolution experiments, we also created a control MDCK cell line expressing DHFR.YFP (MDCK$^{YFP}$). This MDCK$^{YFP}$ cell line does not display TMP-dependent upregulation of HSF1-dependent chaperones, indicating that TMP induces HSF1 activity in our MDCK$^{HSF1}$ cells specifically by stabilizing DHFR.cHSF1 (*Figure 1—figure supplement 1B*).

A second approach to create an altered folding environment is to inhibit individual chaperones. Here, we employed the Hsp90 inhibitor STA-9090, a small molecule capable of targeting multiple Hsp90 isoforms (*Ying et al., 2012*). Notably, Hsp90 inhibition using high concentrations of STA-9090 can cause an undesirable compensatory heat shock response (*Moore et al., 2016*), resulting in activation of HSF1 and upregulation of Hsp70, Hsp40, and other HSF1 targets. Such compensatory

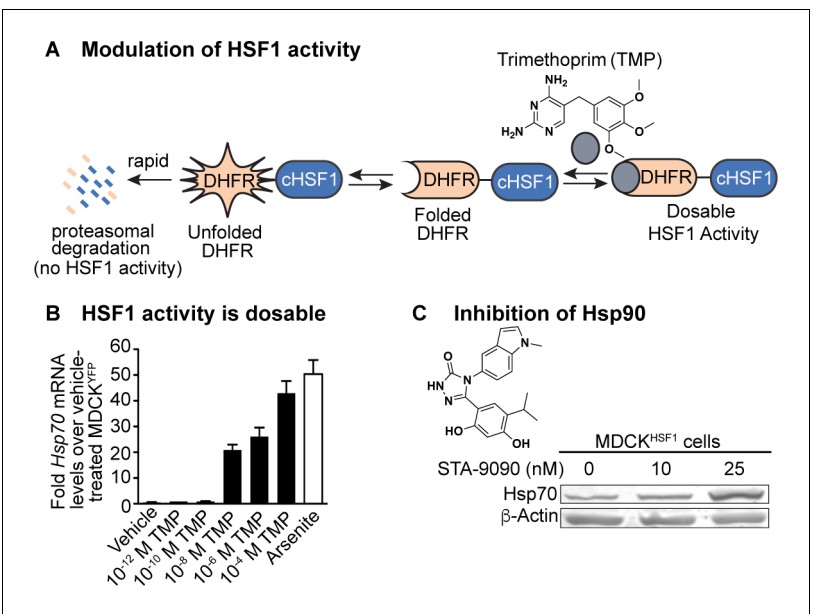

**Figure 1.** Chemical biology methods to modify the host cell's proteostasis environment. (**A**) Destabilized domain technology for stress-independent control of HSF1 activity with trimethoprim (TMP). (**B**) Dosable induction of HSF1 activity by increasing concentrations of TMP shown by increases in *Hsp70* transcripts up to physiologically relevant levels; arsenite is a positive control for endogenous HSF1 activation. Transcript levels normalized to vehicle-treated MDCK[YFP] cells; error bars represent SEM between biological triplicates. (**C**) 10 nM STA-9090 does not induce a compensatory heat shock response (representative blot shown; N = 3). *Figure 1—figure supplement 1*. Validation of chemical biology tools used to perturb proteostasis. *Figure 1—figure supplement 2*. Heat shock protein transcript expression during influenza infection in modulated proteostasis environments.
DOI: https://doi.org/10.7554/eLife.28652.003

The following figure supplements are available for figure 1:

**Figure supplement 1.** Validation of chemical biology tools used to perturb proteostasis.
DOI: https://doi.org/10.7554/eLife.28652.004

**Figure supplement 2.** Heat shock protein transcript expression during influenza infection in modulated proteostasis environments.
DOI: https://doi.org/10.7554/eLife.28652.005

HSF1 activation would convolute interpretation of our results. Thus, we treated with the highest possible STA-9090 concentration that does not induce a compensatory heat shock response in MDCK cells. We selected the concentration used via a functional assay for all experiments, examining Hsp70 protein levels by immunoblotting to ensure the absence of any heat shock response signature (*Figure 1C*). To confirm that we are still engaging Hsp90 at the low STA-9090 concentration used in our serial passaging experiments, we performed a cellular thermal shift assay (*Martinez Molina et al., 2013*) and observed a small but highly reproducible increase in Hsp90 thermal stability upon STA-9090 treatment (*Figure 1—figure supplement 1D*).

Because Hsp90 inhibition can be employed in our MDCK[HSF1] cell line, these methods access three distinctive host proteostasis environments in a single cell line dependent only on small molecule treatment. The use of just a single cell line for experiments (MDCK[HSF1]) and for controls (MDCK[YFP]) minimizes any possible cell line-dependent bias in our evolution experiments. Importantly, monitoring chaperone levels in the context of influenza A/Wuhan/1995 (H3N2) infection shows that, under our infection conditions, influenza itself neither induces heat shock protein transcripts nor interferes with our method to activate HSF1. Thus, we have full user control of the host proteostasis environment during a progressing infection (*Figure 1—figure supplement 2*).

To further characterize these perturbed host environments, we performed RNA-Seq for each treatment in the MDCK[HSF1] and MDCK[YFP] cell lines. We observed a total of only 118 transcripts whose expression is altered ≥2 fold with a *p*-value <$10^{-5}$ in any one or more the treatments employed, indicating that we are remodeling only a small portion of the transcriptome

(comprehensive quality control and RNA-Seq results are provided in *Figure 2—source data 1–2*). A heat map for these 118 genes highlights that two distinctive cellular environments are indeed created by HSF1 activation and Hsp90 inhibition in MDCK$^{HSF1}$ cells (*Figure 2A*). Moreover, only two transcripts meet these cutoffs upon TMP treatment in MDCK$^{YFP}$ cells, confirming that the remodeled proteostasis environment upon TMP treatment of MDCK$^{HSF1}$ cells is specifically due to HSF1 activation.

The volcano plots in *Figure 2B* show the distribution of differentially expressed genes for HSF1 activation and Hsp90 inhibition in MDCK$^{HSF1}$ cells, relative to the basal environment, with selected transcripts labeled (for a list of all transcripts meeting these thresholds see *Figure 2—source data 3*). As expected, stress-independent HSF1 activation upregulates numerous classic heat shock response genes (*Ryno et al., 2014*), including *HSP70*, *BAG3*, *HSP90AA1*, *DNAJB1*, and *FKBP4* (*Figure 2B*). This remodeling of the cellular proteostasis network is limited to nuclear and cytosolic proteostasis mechanisms, as ER proteostasis network components are not induced by TMP treatment of the MDCK$^{HSF1}$ cells (*Figure 2C*). Moreover, transcript-level proteostasis network remodeling is not observed upon STA-9090 treatment, with the exception of a 2.2-fold induction of *HSPB1*, as expected given the low concentration of inhibitor employed. We observe modest upregulation of several transcripts involved in transcription factor regulation and DNA damage upon HSF1 activation, such as *CAMK1*, *RELB*, and *PARP3*, consistent with the intimate role of HSF1 in cytoprotection (*Morimoto and Santoro, 1998*). STA-9090 slightly upregulates several interferon response-related transcripts, including *OAS2* and *MX2* (2.9 and 2.2 fold-change, respectively), a phenomenon that has been previously reported upon treatment with other Hsp90 inhibitors (*Yang et al., 2006*; *Donzé et al., 2001*; *Shang and Tomasi, 2006*). In summary, our RNA-Seq data are fully consistent with the chemically-controlled creation of three unique host cell proteostasis environments in MDCK$^{HSF1}$ cells.

## Serial passaging to emulate influenza evolution

We next serially passaged influenza A/Wuhan/1995 in our three distinctive, small molecule-controlled host environments to test the hypothesis that host proteostasis impacts influenza evolution. Prior to each infection, we split a population of MDCK$^{HSF1}$ cells and treated with TMP to activate HSF1, STA-9090 to inhibit Hsp90, or vehicle (*Figure 3A*). We infected at a low multiplicity of infection (MOI), ranging from 0.001 to 0.04 infectious virions/cell, to minimize non-viable variants hitchhiking with functional variants owing to co-infection of a single cell (*Figure 3—figure supplement 1A*). We performed 23 serial passages in biological triplicate in each proteostasis environment. We also performed identical passaging experiments ± TMP in our MDCK$^{YFP}$ cells to control for any possible off-target effects of TMP treatment or expression of a DHFR-fusion protein in our evolution experiments (*Figure 3B*).

Following each passage, clarified viral supernatant was harvested for hemagglutination titering and subsequent infection to initiate the next passage (*Figure 3C*, *Figure 3—figure supplement 1B*). Importantly, infectious titering at intermittent passages confirmed that the MOI did not vary systematically between the three distinctive proteostasis environments (*Figure 3—figure supplement 1A*). Thus, differences observed in evolutionary trajectories cannot be attributed to either systematically altered rates of viral growth in the three different host proteostasis environments studied here or to gross differences in cell health.

Next, we sought to evaluate whether our serial passaging strategy provides a valid platform for modeling influenza evolution. In the absence of a strong exogenous selection pressure, such as an antiviral drug or antibody, we would predict that the influenza genome experiences purifying selection, meaning that most amino acid substitutions result in reduced fitness relative to the wild-type consensus sequence. We assessed the extent of purifying selection by determining the ratio of non-synonymous to synonymous substitutions at each passage, normalized to the ratio of non-synonymous sites to synonymous sites in the influenza genome (3.5: 1), defined as $D_n/D_s$. As expected, non-synonymous mutations are selected against in all folding environments, as indicated by a $D_n/D_s < 1$ throughout our serial passaging (*Figure 3D*) (*Breen et al., 2012*).

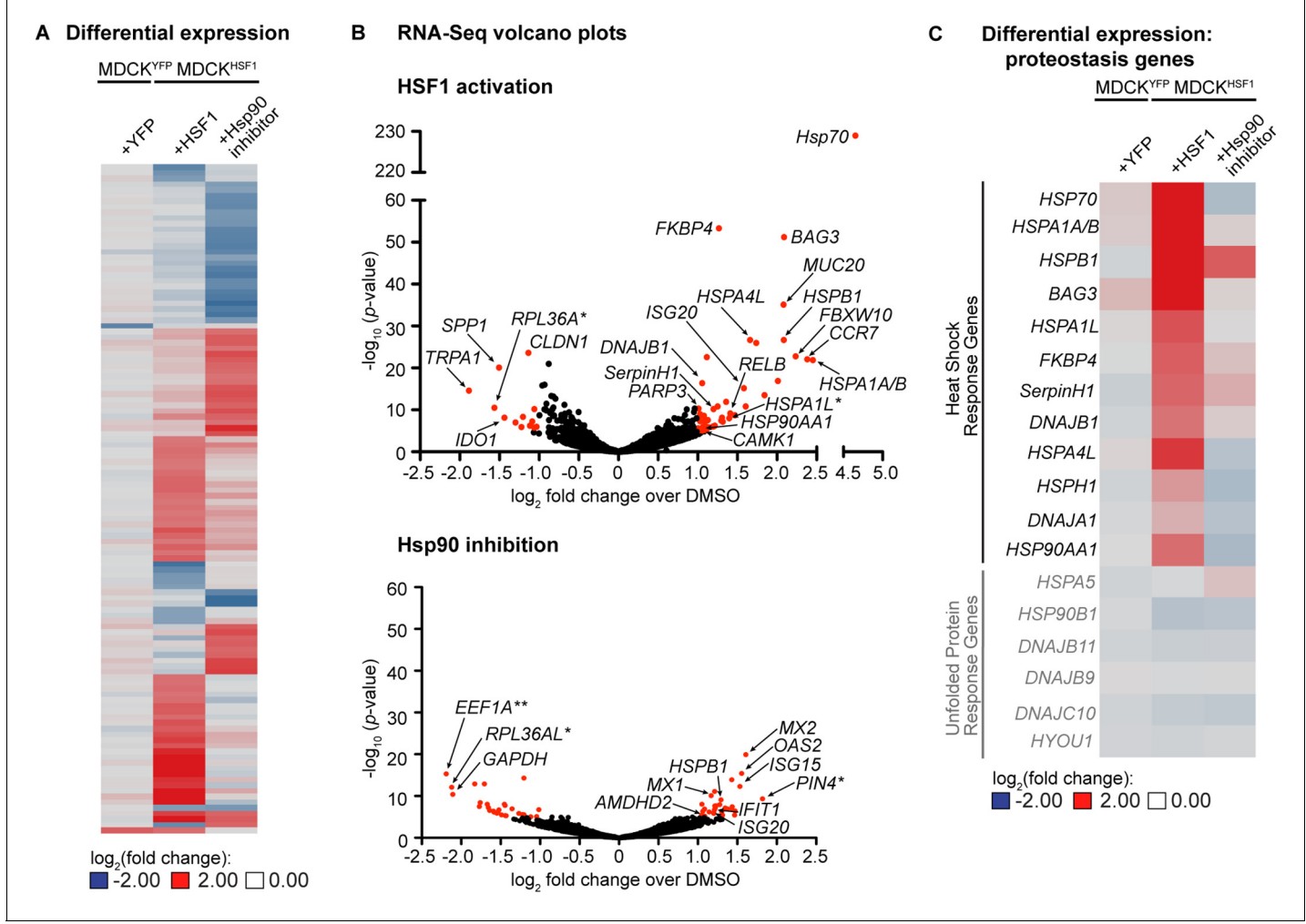

**Figure 2.** Transcriptomic analysis of perturbed host cell proteostasis environments. (A) Differential expression analysis of MDCK[HSF1] cells treated for 24 hr with 10 µM TMP (+HSF1) or 10 nM STA-9090 (+Hsp90 inhibitor) and MDCK[YFP] cells treated for 24 hr with 10 µM TMP (+YFP), normalized to vehicle treatment in the corresponding cell line. Transcripts displaying ≥2 fold changes in expression with $p$-values $<10^{-5}$ for any of the treatments are included in the heat map (118 transcripts total). (B) Volcano plots showing the global distribution of expressed transcripts upon HSF1 activation or Hsp90 inhibition as in *Figure 2A*. Transcripts displaying ≥2 fold changes in expression with $p$-values $<10^{-5}$ are shaded red. Outliers and transcripts encoding proteostasis network components, stress response genes, and transcription factor modulators are labeled. *=unannotated canine genes homologous to the indicated gene across multiple species; **=unannotated canine genes that fell within the indicated paralog gene family with partial homology. (C) Heat map showing the differential expression of heat shock response and unfolded protein response gene transcripts upon HSF1 activation or Hsp90 inhibition as in *Figure 2A*. *Figure 2—source data 1*. RNA-Seq characterization of MDCK[HSF1] and MDCK[YFP] cells: quality control metrics. *Figure 2—source data 2*. RNA-Seq characterization of MDCK[HSF1] and MDCK[YFP] cells: differential transcript expression. *Figure 2—source data 3*. RNA-Seq characterization of MDCK[HSF1] and MDCK[YFP] cells: list of all transcripts displaying ≥2 fold changes in expression with $p$-values $<10^{-5}$ for each treatment.

DOI: https://doi.org/10.7554/eLife.28652.006

The following source data is available for figure 2:

**Source data 1.** RNA-Seq characterization of MDCK[HSF1] and MDCK[YFP] cells: quality control metrics.

DOI: https://doi.org/10.7554/eLife.28652.007

**Source data 2.** RNA-Seq characterization of MDCK[HSF1] and MDCK[YFP] cells: complete RNA-Seq datasets upon vehicle treatment, HSF1 activation, or Hsp90 inhibition in MDCK[HSF1] cells and vehicle treatment or YFP activation in MDCK[YFP] cells.

DOI: https://doi.org/10.7554/eLife.28652.008

**Source data 3.** RNA-Seq characterization of MDCK[HSF1] and MDCK[YFP] cells: list of all transcripts displaying ≥2 fold changes in expression with $p$-values $<10^{-5}$ for each treatment.

DOI: https://doi.org/10.7554/eLife.28652.009

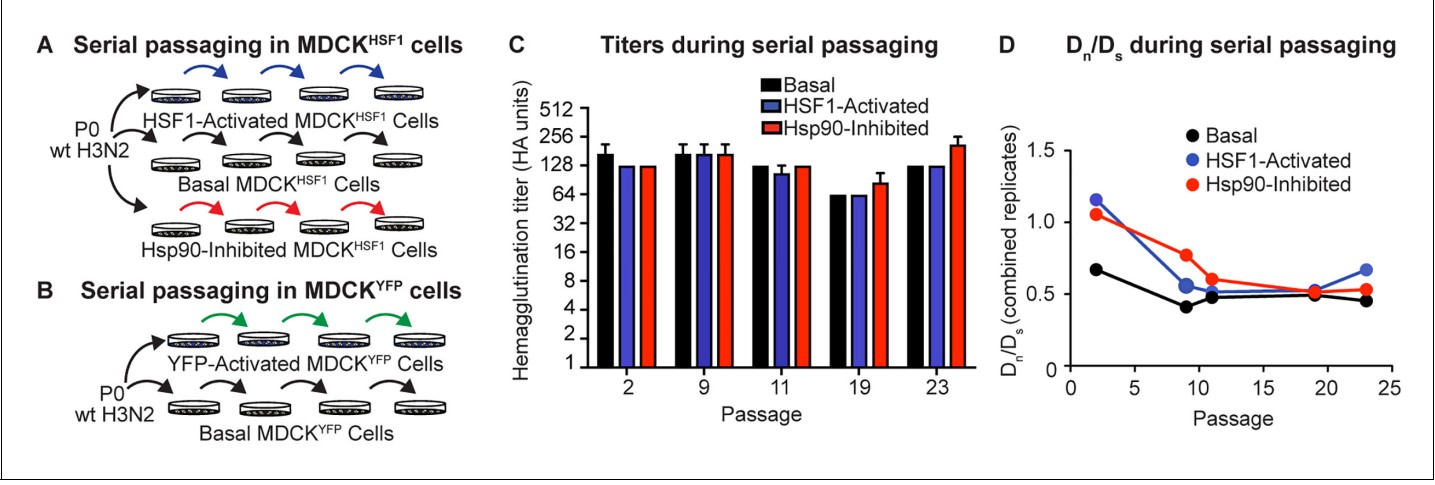

**Figure 3.** Serial passaging of Influenza A/Wuhan/1995 H3N2. (**A**) Serial passaging workflow in modified proteostasis environments in MDCK[HSF1] cells. (**B**) Serial passaging workflow in control MDCK[YFP] cells. (**C**) Hemagglutination titers at intermittent passages for each folding environment; error bars represent SEM for biological triplicates. (**D**) $D_n/D_s$ ratios for each viral population, normalized to the ratio of non-synonymous sites to synonymous sites in the influenza genome (3.5). *Figure 3—figure supplement 1*. Multiplicity of infection and hemagglutination titers during serial passaging.
DOI: https://doi.org/10.7554/eLife.28652.010

The following figure supplement is available for figure 3:

**Figure supplement 1.** Multiplicity of infection and hemagglutination titers during serial passaging.
DOI: https://doi.org/10.7554/eLife.28652.011

## Nature of selection pressure differs in modified proteostasis environments

To comparatively evaluate the selection pressure placed on influenza by our three distinctive host proteostasis environments, we constructed variant frequency distributions (site frequency spectra; SFS) for each viral population (*Nielsen, 2005*). In the SFS (*Figure 4A–B*), we separate mutations present above our sequencing error threshold (1.5%) in a given viral population into either non-synonymous or synonymous groups. The bars in the SFS charts represent the portion of variants in a viral population that fall within a given frequency bin, averaged across biological triplicates. As expected, the resulting SFS for non-synonymous mutations (*Figure 4A*) show that early passages consist entirely of low frequency variants. As the passaging experiment progresses, the high mutation rate of influenza maintains a substantial population of low frequency variants. However, the proportion of low frequency variants in the viral population decreases as selected mutations increase in frequency and become fixed. This phenomenon is highlighted in later passages as variants begin to occupy high frequency bins and a bimodal, U-shaped distribution emerges.

The time resolution afforded by our serial passaging and sequencing strategy provides the opportunity to assess the strength of selection pressure on the influenza genome. Non-synonymous mutations fix latest in the Hsp90-inhibited environment and earliest in the HSF1-activated environment (*Figure 4A*; outlined in green). For instance, in the Hsp90-inhibited environment, no mutations have exceeded 60% frequency by passage 9 or 80% frequency by passage 11. In contrast, in the HSF1-activated environment, mutations exceed 80% frequency as early as passage 9 and become fixed (>90%) by passage 11. The basal environment lies between these two extremes, with mutations exceeding only 70% frequency by passage 9 and with no mutations yet being fixed at passage 11.

As mutations in each environment increase in frequency, all the distributions achieve similar U-shaped, bimodal distributions by passage 23. For the basal and HSF1-activated environments, the distributions become U-shaped by passage 11, whereas for the Hsp90-inhibited environment, this distribution emerges only after passage 19. To assess the significance of these differences in the shape of the SFS, we applied the Mann-Whitney test, which is a statistical test for comparing two distributions (*Mann and Whitney, 1947*). Indeed, by passage 11, the first passage at which we observe fixed mutations in any environment, the shape of the non-synonymous SFS for influenza

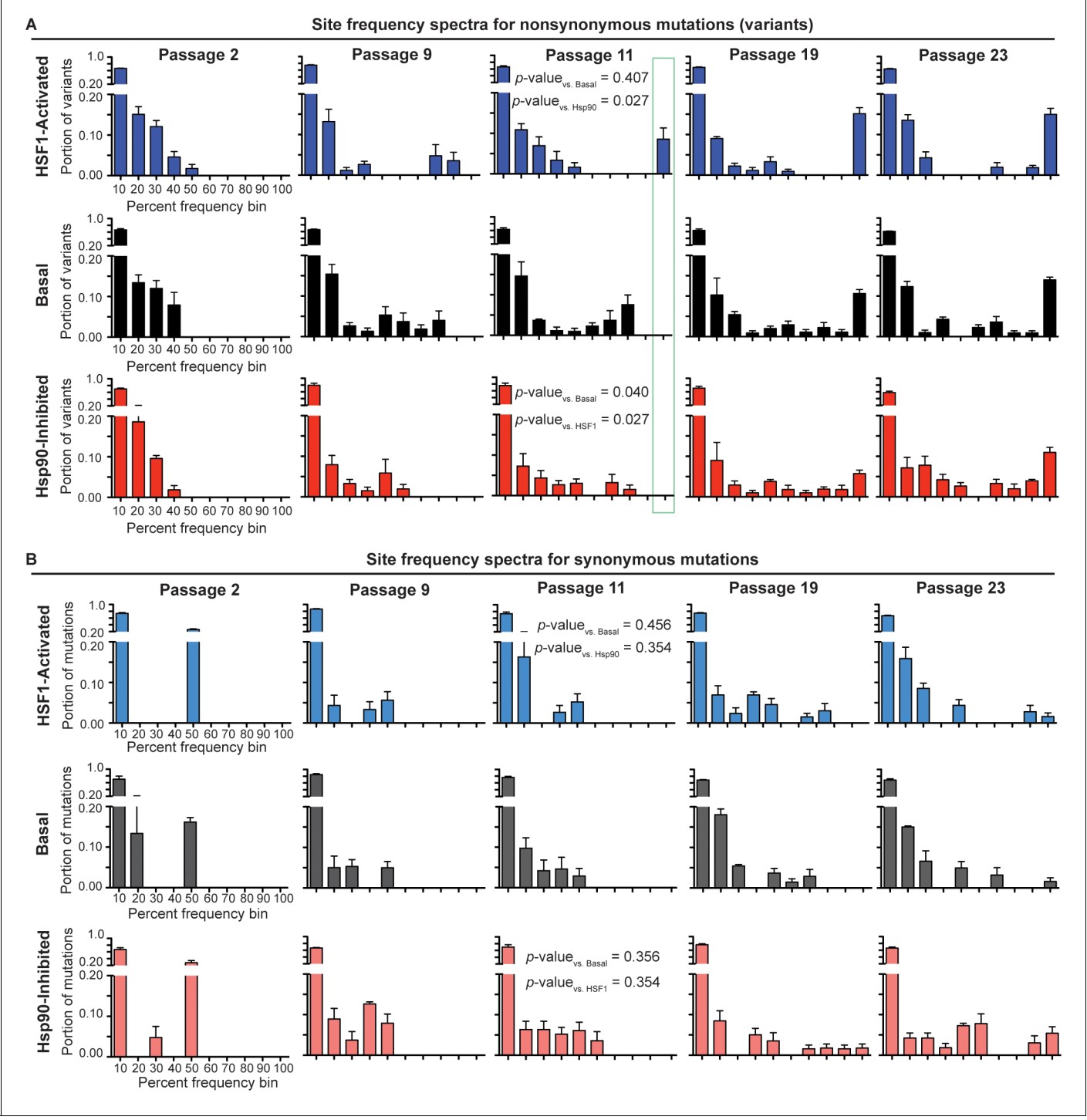

**Figure 4.** Site frequency spectra show frequency distribution of non-synonymous (**A**) and synonymous (**B**) mutations in a given folding environment at a particular passage. Average between biological triplicates is plotted; error bars represent SEM. The Mann-Whitney test was used to compare the distributions of the SFS and assess statistical significance of these differences; resulting *p*-values are shown for passage 11 SFS. *Figure 4—figure supplement 1*. Trajectories for non-synonymous and synonymous mutations that increase in frequency during serial passaging. *Figure 4—source data 1*. List of ten highest % frequency non-synonymous mutations for each proteostasis environment and passage. *Figure 4—source data 2*. List of ten highest % frequency synonymous mutations for each proteostasis environment and passage.

DOI: https://doi.org/10.7554/eLife.28652.012

The following source data and figure supplement are available for figure 4:

**Source data 1.** List of ten highest % frequency non-synonymous mutations for each proteostasis environment and passage.

*Figure 4 continued on next page*

*Figure 4 continued*

DOI: https://doi.org/10.7554/eLife.28652.014

**Source data 2.** List of ten highest % frequency synonymous mutations for each proteostasis environment and passage.

DOI: https://doi.org/10.7554/eLife.28652.015

**Figure supplement 1.** Trajectories for non-synonymous (**A**) and synonymous (**B**) mutations that increase in frequency during serial passaging.

DOI: https://doi.org/10.7554/eLife.28652.013

evolved in the Hsp90-inhibited environment is significantly different from that of influenza evolved in the basal and HSF1-activated environments (*Figure 4A*). This reduced rate of adaptation in Hsp90-inhibited versus basal and HSF1-activated environments is also evident from plotting individual mutation trajectories (*Figure 4—figure supplement 1*), revealing specific mutations that increase in frequency more gradually when Hsp90 is inhibited. In contrast, the overall shapes of the basal and HSF1-activated SFS are not significantly different.

As an internal control, we also examined the SFS and mutation trajectories for synonymous variants (*Figure 4B*). We find that synonymous mutations for each environment have similar SFS within each passage, and that the Mann-Whitney test cannot distinguish between the synonymous SFS in different proteostasis environments at any stage of the serial passaging experiment (*Figure 4B*). Moreover, unlike non-synonymous mutations (see below and also see *Figure 4—source data 1*), specific synonymous mutations do not fix reproducibly in one particular environment (*Figure 4—source data 2*). This observation is consistent with our expectation that, although the influenza genome does undergo selection at the RNA level (*Air et al., 1990*), the host proteostasis-based perturbations of HSF1 activation and Hsp90 inhibition should primarily affect influenza evolution by modulating protein-level folding/function. The similar SFS for synonymous mutations also confirm that differences between folding environments for non-synonymous mutations cannot be attributed to altered viral growth rates.

Taken together, these observations draw an intriguing picture of the impact of host proteostasis on viral evolution. On the one hand, there is little effect on strongly deleterious mutations, as illustrated by the similar $D_n/D_s$ ratios across environments (*Figure 3D*). On the other hand, the rate of adaptation by accumulation of advantageous mutations is significantly reduced in the Hsp90-inhibited environment, as compared to the basal and HSF1-activated environments. This signature of altered selection pressure in distinctive host proteostasis environments highlights the importance of this under-appreciated factor impacting viral evolution.

## Influenza protein mutational landscapes are modulated by host proteostasis

The data in *Figure 4* show that the host proteostasis environment is a determinant of the nature of selection pressure placed on the influenza genome. Next, we examined how the mutational landscapes of individual influenza proteins are influenced by these same environments. We anticipated perturbations of HSF1- and Hsp90-mediated proteostasis mechanisms might affect the mutational landscape of polymerase subunits, as their interactions with cytosolic chaperones are well-established (*Momose et al., 2002*; *Naito et al., 2007*; *Chase et al., 2008*). Moreover, our sequencing coverage of the polymerase subunits PA and PB1 was sufficient to rigorously analyze the distribution of mutations in those two genes. We aligned all non-synonymous variants present in passage 23 to the secondary structure, the amino acid relative surface accessibility, and the amino acid conservation score (or site entropy; *Figure 5A–B*). Using this analysis, we identified regions of PA and PB1 (outlined in *Figure 5A–B*) with a high density of non-synonymous, but not synonymous, mutations with frequencies above our sequencing error threshold (1.5%). We termed these regions mutational hotspots.

Most mutational hotspots appear in all host environments, indicating sites under strong positive selection in this experimental setting and/or suggesting inherent mutational tolerance of those protein regions that is independent of host proteostasis. For example, in polymerase PA, residues 590–640 comprise a mutational hotspot present in every host environment (*Figure 5A*). The region is surface-exposed prior to polymerase complex assembly (*Figure 5A*, relative surface accessibility; RSA) and the mutations largely occur at sites known to be hypervariable (*Figure 5A*, site entropy; SE) (*Pei and Grishin, 2001*; *Bao et al., 2008*). In contrast, certain other mutational hotspots are specific

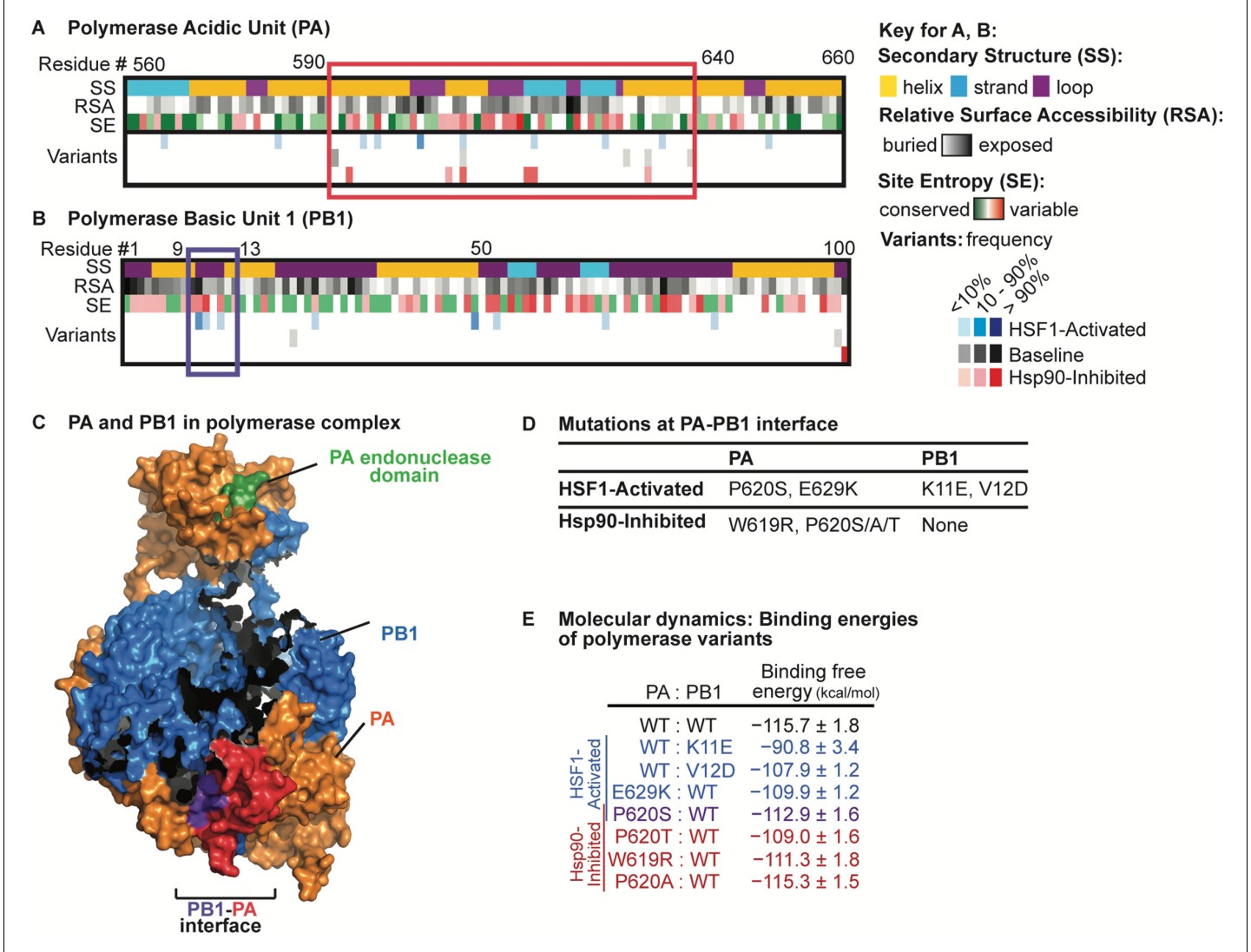

**Figure 5.** Analysis of non-synonymous mutations observed in distinctive proteostasis environments. Aligned variants were observed in any of three biological replicates. (A) Alignment of PA variants to secondary structure, relative surface accessibility, and site entropy. Residues 560–660 are shown, highlighting a mutational hotspot that occurs in all proteostasis environments outlined in red. (B) Alignment of PB1 variants to secondary structure, relative surface accessibility, and site entropy. Residues 1–100 are shown, highlighting a mutational hotspot observed only when HSF1 is activated outlined in purple. (C) Mutational hotspots mapped onto the PA-PB1 complex crystal structure (PDBID 4WSB) (*Reich et al., 2014*). PA hotspot is shaded red; PB1 hotspot is shaded purple. (D) List of amino acid substitutions likely to affect PA-PB1 binding that appear in the HSF1-activated and/or Hsp90-inhibited environments. (E) Binding free energy* of the $PA_C$–$PB1_N$ complex (kcal/mol; calculated as the contribution ($PA_C$–$PB1_N$ complex) – contribution ($PA_C$) – contribution ($PB1_N$)). Binding free energies shown for simulations with wild-type and mutant subunits; reported error is SEM. *Excludes contribution from solute configuration entropy. *Figure 5—source data 1*. Energy contributions from molecular dynamics simulations.
DOI: https://doi.org/10.7554/eLife.28652.016

The following source data is available for figure 5:

**Source data 1.** Molecular Dynamics Simulations: Energy Contributions.
DOI: https://doi.org/10.7554/eLife.28652.017

to or absent from one or more host proteostasis environments. For example, we observe eight mutations in the N-terminal 100 residues of PB1 in the HSF1-activated environment, two mutations in the baseline environment, and only one mutation in the Hsp90-inhibited environment (*Figure 5B*).

The identification of mutational hotspots in the N-terminal domain of PB1 and residues 590–640 of PA is interesting, as their interaction is required for assembly of the mature polymerase complex

(*Figure 5C*) (*Reich et al., 2014*; *Perez and Donis, 2001*; *Obayashi et al., 2008*). In particular, residues 9–13 of PB1 span the loop portion of a helix-loop-helix motif (*Figure 5B*, secondary structure; SS) that critically defines the interface (*Figure 5C*, purple) with PA (*Figure 5C*, red) (*Reich et al., 2014*; *Perez and Donis, 2001*; *Obayashi et al., 2008*). Our observation that amino acid substitutions in PA at the interface are tolerated in all host environments while substitutions in PB1 at the interface are tolerated only when HSF1 is activated (*Figure 5A–B*) prompted us to further investigate the consequences of these amino acid substitutions. First, we selected mutations that alter amino acid residues known to make direct contacts at the PA–PB1 interface (*Liu and Yao, 2010*) and that exceed 1.5% frequency (our sequencing error threshold) for at least two passages (*Figure 5D*). To estimate their effects on PA:PB1 complex stability, we performed molecular dynamics simulations between wild-type and mutant forms of $PA_C$ (specifically, a C-terminal PA domain covering residues 257–716) and $PB1_N$ (an N-terminal PB1 domain spanning the first 15 residues) (*Liu and Yao, 2010*). We observed that the PA variants, which occur in both the HSF1-activated and Hsp90-inhibited environments, have slightly less favorable binding energy with wild-type $PB1_N$ (*Figure 5E*). The PB1 variants, which occur exclusively in the HSF1-activated environment, are also destabilizing, with the K11E substitution very strongly destabilizing the complex (*Figure 5E*).

These data suggest that, whereas our perturbations of host proteostasis may be necessary to accommodate strongly destabilizing mutations in PB1, they may not be important for moderately destabilizing variants in PA. We note that co-occurrence of certain PA variants (e.g., E629K in PA) can compensate for the strongly destabilizing effects of observed PB1 substitutions (e.g., the combination of E629K in PA and K11E in PB1 results in a stable complex; *Figure 5—source data 1*). However, the mutation frequencies are not strongly correlated in our data and likely occur in distinctive influenza genomes. Cumulatively, these observations indicate that host proteostasis impacts the fitness of destabilizing amino acid substitutions in influenza proteins in a protein-specific manner.

## Divergent mutational trajectories in HSF1-activated versus Hsp90-inhibited environments

Next, we analyzed the mutational trajectories of variants present at high frequency in our founder virus to identify those with significantly altered fitness between proteostasis environments. Such variants had the opportunity to be selected for or against in all environments and thus are less subject to the stochasticity inherent in a serial passaging-based evolution experiment. As we would expect, some non-synonymous variants (e.g., HA N262K, NS1 F150S, and PB2 F14S) follow similar trajectories regardless of host proteostasis (*Figure 4—source data 1*). Much more interesting are the HA and PA variants that exhibit divergent trajectories upon serial passaging in distinct folding environments (*Figure 6A–B* and *Figure 4—source data 1*). Notably, we do not observe any synonymous variants with consistently divergent trajectories between environments (*Figure 4—source data 2*) indicating that the divergent PA and HA trajectories are in fact due to selection at the protein level.

In HA, three variants (N144T, V242I, and M246I) present at 20% frequency in our founder virus become fixed by passage 19 in all replicates of the HSF1-activated and basal environments, but in only one replicate of the Hsp90-inhibited environment (*Figure 6A*). Moreover, these variants completely fall out of two replicates of the Hsp90-inhibited environment by passage 9. All three mutations share identical mutational trajectories and V242I and M246I occur together in sequencing reads. Hence, the trajectory of only one variant, N144T, is shown in *Figure 6A*. The three mutations occur in the globular head domain of HA that binds cellular sialic acid and has inherently high mutational tolerance (*Thyagarajan and Bloom, 2014*). These mutations may increase the affinity of HA for MDCK sialic acid receptors, but may be less fit when Hsp90 is inhibited. HA folding occurs in the endoplasmic reticulum, which contains an Hsp90 isoform (Grp94) that is also inhibited by STA-9090 (*Marzec et al., 2012*). In the absence of active Grp94, these HA variants may have compromised folding and/or intracellular trafficking, resulting in diminished fitness relative to wild-type HA regardless of any functional advantage they may confer in other host proteostasis environments.

In PA, we observed that the H452Q variant is present at 20% frequency in the founder virus and becomes fixed in two replicates of the Hsp90-inhibited environment (*Figure 6B*). This variant does not become fixed in the basal environment, and is selected against in the HSF1-activated environment. Interestingly, although the H452Q mutation is far from the PA endonuclease site, (*Reich et al., 2014*) it is known to occur preferentially in reassorted viruses compared to pure strains, which is indicative of a fitness cost (*Zeldovich et al., 2015*).

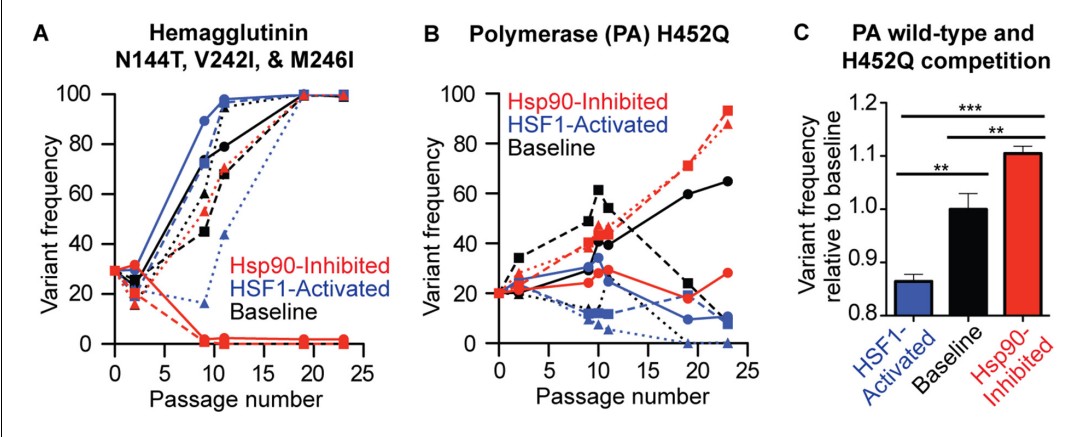

**Figure 6.** HA and PA display divergent mutational trajectories in HSF1-activated versus basal versus Hsp90-inhibited environments. (A) Mutational trajectories of HA variants for each biological replicate in all three proteostasis environments; note that the N144T, V242I, and M246I variants all have identical trajectories. (B) Mutational trajectories of the PA H425Q variant for each biological replicate in all three proteostasis environments. (C) Reverse genetics competition between H452Q and wild-type PA in each proteostasis environment. Average variant allele frequency was normalized to that in the basal proteostasis environment. SEM for biological triplicates is shown. Statistical significance was calculated using a Student's *t*-test.
DOI: https://doi.org/10.7554/eLife.28652.018

To unequivocally determine the fitness of the H452Q variant relative to wild-type PA in HSF1-activated and Hsp90-inhibited host cells, we performed reverse genetics to enable head-to-head competition of the variants, thereby quantitatively establishing their relative fitness in our three distinct host proteostasis environments. We prepared wild-type and H452Q PA-containing influenza populations that were otherwise genetically identical. Next, we co-infected host cells displaying HSF1-activated, basal, or Hsp90-inhibited folding environments with equivalent amounts of each virus at low MOI and sequenced the resulting populations after the competition to quantify variant fitness relative to wild-type. We found that Hsp90 inhibition does indeed significantly enhance the fitness of the H452Q variant relative to wild-type PA, while HSF1 activation significantly reduces it (*Figure 6C*). One likely possibility is that Hsp90 delays subunit assembly or directs destabilized PA variants to degradation, as H452Q has a predicted ΔΔG of +1.10 kcal/mol relative to wild-type PA (*Yin et al., 2007*). In either scenario, this observation that the fitness of an otherwise deleterious amino acid substitution in a non-autonomous Hsp90 client can be enhanced by chaperone inhibition but reduced by HSF1 activation is provocative.

## Discussion

Considerable evidence suggests that autonomous chaperone networks can critically influence the evolution of their endogenous client protein partners (*Cowen and Lindquist, 2005*; *Queitsch et al., 2002*; *Lachowiec et al., 2015*; *Sangster et al., 2007*, *2008a*, *2008b*; *Rohner et al., 2013*; *Whitesell et al., 2014*; *Geiler-Samerotte et al., 2016*; *Rutherford and Lindquist, 1998*; *Tokuriki and Tawfik, 2009b*). However, prior to this work, the possibility that host chaperones significantly modulate pathogen evolution had not been rigorously investigated. Moreover, in eukaryotic systems, research has focused largely on Hsp90 inhibition. Here, we not only define a new role for host proteostasis in influenza evolution, but we also show that two unique proteostasis perturbations, HSF1 activation and Hsp90 inhibition, have distinctive consequences for client protein evolution. These consequences are revealed at the levels of the whole genome, individual genes, and specific mutations.

At the whole genome-level, our data indicate that non-synonymous mutations are fixed more slowly when Hsp90 is inhibited and more quickly when HSF1 is activated. Moreover, the overall shape of the Hsp90-inhibited passage 11 SFS is significantly different from that observed in the other two environments. Changes in the rate of adaptation could be caused by buffering of mildly deleterious mutations or by potentiation of advantageous variants (*Cowen and Lindquist, 2005*;

*Queitsch et al., 2002*; *Lachowiec et al., 2015*; *Sangster et al., 2007, 2008a, 2008b*; *Rohner et al., 2013*; *Whitesell et al., 2014*; *Geiler-Samerotte et al., 2016*; *Rutherford and Lindquist, 1998*; *Cetinbaş and Shakhnovich, 2013*). In the case of buffering, the rate of fixation of advantageous variants (driver mutations) would be decreased if the effects of mildly deleterious variants linked with the driver (passenger mutations) are rendered more damaging by Hsp90 inhibition (*McFarland et al., 2013, 2014*). In the case of potentiation, Hsp90 activity could necessitate emergence of the observed high frequency variants. Alternatively, Hsp90 activity may reduce weakly deleterious effects of driver mutations on their carrier proteins (e.g., disrupted folding caused by an otherwise beneficial mutation), thereby increasing the selective advantage provided by the drivers. Though we cannot yet fully resolve which of these factors, or combinations thereof, are at play here, overall, we observe that host proteostasis indeed modulates the pace of influenza evolution.

At the level of individual genes, we find that mutational tolerance at the interface of the PA:PB1 influenza polymerase complex, a region essential for polymerase assembly and activity, is impacted by host proteostasis. Interestingly, moderately destabilizing variants in PA are tolerated in all three host proteostasis environments studied. In contrast, amino acid substitutions in the N-terminus of PB1 are observed only when HSF1 is activated. These PB1 variants strongly destabilize the PB1:PA complex and thus may be accessible only in host cells with high chaperone levels. Indeed, polymerase assembly is known to be mediated by cytosolic host chaperones (*Naito et al., 2007*). These observations indicate that the evolution of the influenza polymerase complex may be modulated by host proteostasis in significant ways.

At the level of individual mutations, our data demonstrate that host proteostasis impacts the fitness of specific mutations in at least two viral proteins. Intriguingly, the directionality of this effect seems to be specific to a given variant or protein. For example, the apparently destabilizing H452Q PA variant is significantly more fit when Hsp90 is inhibited, but significantly less fit when HSF1 is activated. These results highlight that the impact of proteostasis perturbation on evolution is currently difficult to predict *a priori* and demands further study, as one might by default assume that chaperone inhibition would enhance the fitness costs of such a mutation. We also identified variants in HA that behave in the opposite manner, displaying apparently enhanced fitness when HSF1 is activated. These divergent effects of host proteostasis on the fitness of individual variants illustrate the complexity of the interactions between viral proteins and host proteostasis mechanisms.

Thus, we observe that host proteostasis modulates the nature of selection on the influenza genome, the mutational tolerance of specific influenza proteins, and the trajectories of particular variants. These results are especially compelling because consequences of altered host proteostasis for influenza evolution are emerging in the course of a relatively short-timescale evolution experiment, and without exerting additional selection pressure such as treatment with antiviral drugs. As in previous studies on the role of Hsp90 in the evolution of their endogenous protein clients, our observations may derive from direct interactions between host proteostasis components and influenza proteins, or from indirect consequences of perturbing proteostasis. Untangling these possibilities will require detailed biophysical and mechanistic studies.

Regardless of its precise origin, this role for host proteostasis in modulating both the pace and the directionality of influenza evolution is provocative. Our observations raise a number of intriguing questions for future work. How do host proteostasis mechanisms beyond the heat shock response modulate influenza evolution, and what specific chaperones beyond Hsp90 are involved? Can we quantitatively evaluate the magnitude and ultimately predict the directionality of these effects? Are these effects direct consequences of host chaperones engaging influenza clients? If not, what are the mediators? How does host proteostasis impact the evolution of rapidly evolving viruses beyond just influenza? Do physiological states involving altered or perturbed proteostasis (e.g., fever or host-switching) impact viral evolution? Defining the molecular details of this interplay between host proteostasis and viral evolution will be essential to fully elucidate the factors potentiating and constraining viral evolution. Moreover, such studies may eventually enable design of improved antiviral therapeutics and antibodies that are refractory to the evolution of resistance.

## Materials and methods

### Cell culture

MDCK cells were generously provided by Prof. Jianzhu Chen (MIT), and were originally purchased from American Type Culture Collection (Manassas, VA). The identity of these cells was authenticated by STR profiling. MDCK cells were cultured at 37°C in a 5% $CO_2$ atmosphere in DMEM (CellGro) supplemented with 10% fetal bovine serum (CellGro) and 1% penicillin/streptomycin/glutamine (CellGro). Cells were transduced with lentiviruses encoding either the DHFR.HSF1(Δ186–202) or DHFR.YFP gene. Heterostable cells expressing the construct of interest were then selected using 4 µg/mL puromycin. Single colonies were generated by diluting cells to ~40 cells per 96-well plate, expanding the resulting colonies, and functionally testing by qPCR or fluorescence microscopy in the presence or absence of trimethoprim (TMP; 10 µM). All cell lines were periodically tested for myco-plasma using the MycoSensor PCR Assay Kit from Agilent (302109).

### Influenza virus

All experiments were performed with influenza A/Wuhan/1995 (H3N2), which was generously pro-vided by Prof. Jianzhu Chen (MIT).

### Compounds and antibodies

STA-9090 was purchased from MedChem Express, sodium arsenite 0.1 N standardized solution was purchased from Alfa Aesar, TPCK-trypsin was purchased from Sigma Aldrich, TMP was purchased from Alpha Aesar. Mouse monoclonal anti-β-actin was obtained from Sigma (A1978). Rabbit poly-clonal anti-HSP70/72 and rabbit polyclonal anti-Hsp40 antibodies were obtained from Enzo Life Sci-ences (ADI-SPA-811-D and ADI-SPA-400D, respectively). The rabbit monoclonal anti-HSP90 antibody was obtained from Cell Signaling Technologies (C45G5).

### Immunoblotting

MDCK$^{HSF1}$ cells were seeded at 200,000 cells/well in a 6-well plate and treated with 0.01% DMSO, 10 µM TMP, or 10 and 25 nM STA-9090 for 48 hr prior to harvesting cell lysates. 100 µg of protein lysate was separated on a 12% SDS-PAGE polyacrylamide gel, followed by transfer to a nitrocellu-lose membrane. Hsp70, Hsp40, Hsp90, and actin protein levels were determined using the antibod-ies described above. Membranes were incubated with 680 or 800 nm fluorophore-labeled secondary antibodies (LI-COR Biosciences, Lincoln, NE) prior to detection using a LI-COR Biosciences Odyssey Imager. Band intensity quantification was performed using Image J.

### qPCR

MDCK$^{HSF1}$ cells were seeded at 100,000 cells/well in a 12-well plate and treated with 0.01% DMSO, 10 µM TMP, or 10 nM STA-9090 for 24 hr. MDCK$^{YFP}$ cells were treated with 0.01% DMSO or 10 µM TMP for 24 hr, or 100 µM arsenite for 2 hr as a positive control for heat shock response activation. To monitor chaperone levels during influenza infection, MDCK$^{HSF1}$ cells were infected with influenza A/Wuhan/1995 at an MOI of 1 for 8 hr to properly mimic the actual environment of a cell infected with a single influenza virion as in our serial passaging experiment. Cellular RNA was harvested using the Omega RNA Extraction kit with Homogenizer Columns. 1 µg RNA was used to prepare cDNA using random primers (total reaction volume = 20 µL; Applied Biosystems High-Capacity Reverse Transcription kit). The reverse transcription reaction was diluted to 80 µL with water, and 2 µL of each sample was used for qPCR with 2 × Sybr Green (Roche) and primers for canis *RPLP2* (house-keeping gene), *Hsp70*, *Hsp40*, *Hsp90,* and influenza *Matrix* (*Supplementary file 1*). *Hsp* transcript levels were normalized to *RPLP2*. For qPCR of influenza-infected cells, a standard curve was pre-pared with a pDZ plasmid backbone containing the Influenza PR8 M segment to determine influenza *Matrix* copy number, which was used as a positive control for productive infection.

### RNA-Seq

For the MDCK$^{HSF1}$ cell line characterization (*Figure 2*), MDCK$^{HSF1}$ cells were seeded at 100,000 cells/well in a 12-well plate and treated with 0.01% DMSO, 10 µM TMP, or 10 nM STA-9090 for 24 hr. MDCK$^{YFP}$ cells were treated with 0.01% DMSO or 10 µM TMP. Each treatment was done in

biological triplicate. Cellular RNA was harvested using the Qiagen RNeasy Plus Mini Kit with QIAshredder homogenization columns. RNA-Seq libraries were prepared using the Illumina NeoPrep system and were sequenced on an Illumina HiSeq SE40.

## RNA-Seq analysis

Quality control: Reads were aligned against canFam3 (Sept. 2011) using bwa mem v. 0.7.12-r1039 [RRID:SCR_010910] with flags –t 16 –f. Mapping rates, fraction of multiply-mapping reads, number of unique 20-mers at the 5' end of the reads, insert size distributions and fraction of ribosomal RNAs were calculated using dedicated perl scripts and bedtools v. 2.25.0 [RRID:SCR_006646] (*Quinlan and Hall, 2010*; *Huang et al., 2009*). In addition, each resulting bam file was randomly down-sampled to a million reads, which were aligned against canFam3 and read density across genomic features were estimated for RNA-Seq-specific quality control metrics (*Figure 2—source data 1*).

RNA-Seq mapping and quantitation: Reads were aligned against canFam3/ENSEMBL 86 (*Aken et al., 2017*) annotation using STAR v. 2.5.3a with the following flags -runThreadN 8 –runMode alignReads –outFilterType BySJout –outFilterMultimapNmax 20 –alignSJoverhangMin 8 –alignSJDBoverhangMin 1 –outFilterMismatchNmax 999 –alignIntronMin 10 –alignIntronMax 1000000 –alignMatesGapMax 1000000 –outSAMtype BAM SortedByCoordinate –quantMode TranscriptomeSAM with –genomeDir pointing to a 75nt-junction canFam3 STAR suffix array (*Dobin et al., 2013*). Gene expression was quantitated using RSEM v. 1.3.0 [RRID:SCR_013027] with the following flags for all libraries: rsem-calculate-expression –calc-pme –alignments -p 8 –forward-prob 0 against an annotation matching the STAR SA reference (*Li and Dewey, 2011*). Posterior mean estimates (pme) of counts and estimated RPKM were retrieved.

Differential expression analysis: Treatments were compared against DMSO for MDCK$^{HSF1}$ and MDCK$^{YFP}$ cell lines independently. Briefly, differential expression analysis was performed in the R statistical environment (R v. 3.2.3) using Bioconductor's DESeq 2 package on the protein-coding genes only [RRID:SCR_000154] (*Love et al., 2014*). Dataset parameters were estimated using the estimateSizeFactors(), and estimateDispersions() functions; read counts across conditions were modeled based on a negative binomial distribution and a Wald test was used to test for differential expression (nbinomWaldtest(), all packaged into the DESeq() function), using the treatment type as a contrast. Fold-changes, *p*-values and Benjamin-Hochberg-adjusted *p*-values (BH) were reported for each protein-coding gene (*Figure 2—source data 2*). Transcripts changing ≥2 fold with a p-value<$10^{-5}$ are included in *Figure 2—source data 3*. For the annotation of these transcripts, the reference gene annotation was canFam3/ENSEMBL 86 [RRID:SCR_002344]. Canine genes lacking an official gene symbol were manually annotated by individual inspection of the orthology tracks in the UCSC genome browser and reviewing the orthology and paralogy evidence in the ENSEMBL database (release 89). Presumed genes/gene families were assigned based on the depth of gene model conservation across species and orthologs and paralogs assigned by ENSEMBL. If transcripts not identified in the reference annotation displayed very strong homology across multiple species, transcripts were annotated with a single asterisk '*' or termed 'gene, by homology' in *Figure 2B* and *Figure 2—source data 3*, respectively. Alternatively, if transcripts not identified in the reference annotation fell within paralog gene families with partial homology they were annotated with a double asterisk '**' or termed 'gene-like' in *Figure 2B* and *Figure 2—source data 3*, respectively.

## Cellular thermal shift assay (CETSA)

For each biological replicate, MDCK$^{HSF1}$ cells were seeded at 3,000,000 cells/plate in 15 cm plates and treated with 0.01% DMSO or 10 nM STA-9090 for 4 hr . After drug treatment, cells were harvested by trypsinizing, washed twice with PBS, and resuspended in PBS (with DMSO or STA-9090) at a concentration of 20,000,000 cells/mL. This cell suspension was heated in a thermocycler at a temperature gradient (100 μL per temperature per treatment condition) for 3 min, followed by 3 min at room temperature (*Martinez Molina et al., 2013*). Samples were then lysed in a modified RIPA buffer without SDS. Lysate was then separated by centrifugation and run (in technical triplicate) on an SDS-PAGE gel in reducing loading buffer. Protein bands were transferred to a nitrocellulose membrane, which was probed with an Hsp90 primary antibody (C45G5). Membranes were incubated with 800 nm fluorophore-labeled secondary antibodies (LI-COR Biosciences) prior to detection using

a LI-COR Biosciences Odyssey Imager. Band intensity quantification was performed using Image J, and the signal was normalized to the band intensity at 37°C. Technical replicates were averaged within each biological replicate; biological triplicates were then averaged, with the SEM propagated.

## Serial passaging and hemagglutination-based titering

Serial passaging experiments were performed on a 12-well scale in biological triplicate, at 100,000 cells/well and an MOI of 0.002 virions/cell, as estimated by hemagglutination titering. Cells were pre-treated with TMP or DMSO for 24 hr or with STA-9090 for 90 min prior to influenza infection to establish altered proteostasis environments. All infections were performed in DMEM supplemented with penicillin-streptomycin, glutamine, 1 µg/mL TPCK-trypsin, and the relevant small molecules for modulating proteostasis capacity. Infections were allowed to proceed for 48 hr, after which the viral supernatant was harvested, cleared of cellular debris by centrifugation, and titered using a hemagglutination assay (*Eisfeld et al., 2014*). Viral supernatant was diluted 2-fold across round-bottom 96-well plates with PBS and incubated with human red blood cells ($3.92 \times 10^7$ RBC/mL; Innovative Research) for 30–120 min. Wells displaying agglutination were marked influenza-positive, and the titer was determined based on the lowest dilution that was still influenza-positive. Influenza was serially passaged in both the MDCK$^{HSF1}$ and MDCK$^{YFP}$ cell lines in biological triplicate for each treatment condition (0.01% DMSO, 10 µM TMP, and 10 nM STA-9090) for 23 passages. Titering was performed in technical duplicate for each biological replicate, with minimal variation observed between technical replicates.

## Infectious viral titering via tissue culture infectious dose (TCID$_{50}$) assay

We employed a TCID$_{50}$ assay based on that described by Thyagarajan and Bloom (*Thyagarajan and Bloom, 2014*). Briefly, eight 10-fold dilutions of each virus were prepared in quadruplicate in 96-well plates. 5,000 MDCK$^{HSF1}$ cells were then added to each well and incubated at 37°C for 72 hr, after which the wells were scored for the presence of cytopathic effect. The dilutions of virus displaying cytopathic effect in the MDCK$^{HSF1}$ cells were then used to calculate the TCID$_{50}$/µL using https://github.com/jbloomlab/reedmuenchcalculator as described by Thyagarajan and Bloom, (*Thyagarajan and Bloom, 2014*) where virions/µL = 0.69*TCID$_{50}$/µL.

## Reverse genetics

The H452Q PA variant was introduced by site directed mutagenesis on a wild-type PA pHW2000 reverse genetics plasmid for the influenza A/Wuhan/1995 H3N2 strain (generous gift from Prof. Hui-Ling Yen at Hong Kong University) (*Cheung et al., 2014*). The corresponding mutant and wild-type viruses were made by co-transfection on a co-culture of MDCK and HEK 293T cells, previously described by Hoffman et al (*Hoffmann et al., 2000*). Viruses were titered using a TCID$_{50}$ assay (*Thyagarajan and Bloom, 2014*) to perform competition experiments starting with approximately the same amount of wild-type and mutant virus. Competitions were performed under conditions identical to that of the serial passaging experiments, in biological triplicate. RNA from the P0 and P1 viral supernatant was harvested and prepared for sequencing, as described below.

## Deep sequencing

RNA was extracted from 140 µL influenza supernatant from passages P0, P2, P9, P11, P19 and P23 using the Qiagen RNA Mini Kit and eluted in 40 µL molecular biology grade H$_2$O. dsDNA was made from 2.5 µL template RNA using universal influenza primers as previously described, (*Zhou et al., 2009*) except that the small segments were amplified separately from the polymerase segments and pooled following PCR. The amplicons were separated on a 0.8% agarose analytical gel to verify the presence of each influenza genomic segment (8 total). 1 ng of each sample was prepared using the Illumina NexteraXT Sample Preparation kit, omitting the bead normalization step. The concentration of dsDNA in each sample was quantified by Qubit; samples were pooled in sets of 24 and sequenced on an Illumina MiSeq 300v2 cartridge to obtain $2 \times 150$ base pair paired-end reads. RNA from the reverse genetics competition experiments was sequenced using the same protocol, except primers that specifically amplified ~900 bp of the PA gene spanning the mutation site were used for PCR (*Supplementary file 1*).

## Sequencing data analysis

Sequencing reads were aligned against the influenza A/Wuhan/1995 complete CDS sequence, or the influenza A/Wuhan/1995 PA sequence (for reverse genetics sequencing), using bwa mem 0.7.10-r789 [RRID:SCR_010910]. Allele pileups were generated using samtools v.0.1.19 mpileup [RRID:SCR_002105] with flags -d 10000000 –excl-flags 2052, and allele counts/frequencies were extracted (*Li, 2011*; *Li et al., 2009*). Only positions with greater than 600-fold coverage in all replicates of each sample were included in the analysis. Variant alleles present at greater than 1.5% frequency are included in the site frequency spectra (*Figure 4*), protein alignment (*Figure 5*), and mutational trajectories (*Figure 6*) analyses. This frequency threshold is the lowest mutation frequency at which a mutation can be reliably detected in a sample that is sequenced in technical duplicate, for our specific sequencing method and instrument. A 600-fold coverage threshold requires that we observe such a mutation a minimum of nine times.

All trajectories for mutations that increase in frequency during passaging are included in *Figure 4—figure supplement 1*. Increasing trajectories are those best fitting an increasing exponential model $ae^{bx}$, where $a > 0$ and $b > 0.1$. Best fit was determined by comparing least squares regression value. Selected mutations with divergent trajectories between environments are plotted in *Figure 6*.

Site frequency spectra were generated by binning all mutations meeting our coverage (600-fold) and frequency (>1.5%) thresholds into 10% frequency bins and averaging the portion of mutations within a given frequency bin across biological triplicates. To quantify differences in the passage 11 site frequency spectra, the Mann-Whitney test was performed using Graph Pad Prism software (details in statistics section below).

Alignment analyses were performed by aligning mutations meeting our coverage and frequency thresholds to the corresponding secondary structure and relative surface accessibility using DSSP, (*Kabsch and Sander, 1983*; *Joosten et al., 2011*; *Bloom, 2014*) as well as to the site entropy, which was computed using all full-length protein sequences from the Influenza Virus Resource for all Influenza A PA and PB1 sequences except bat influenza sequences (*Pei and Grishin, 2001*; *Bao et al., 2008*; *Katoh and Standley, 2013*). Observed mutations were also mapped onto the corresponding protein crystal structure (PA-PB1 PBDID 4WSB) (*Reich et al., 2014*). Mutational hotspots were manually determined as regions with a high density of exclusively non-synonymous mutations.

Sequencing reads from reverse genetics competition experiments were aligned to the PA reference sequence to determine %-frequency of mutant and wild-type alleles. The ratio of mutant to wild-type PA was calculated for each replicate of each proteostasis environment; the resulting ratio was normalized to the average ratio of mutant to wild-type PA in the basal proteostasis environment. Ratios were then averaged for each set of biological replicates and plotted as a bar chart with SEM. An unpaired *t*-test was used to assess statistical significance between host environments. Mutant protein stability predictions were made using Eris, (*Yin et al., 2007*) employing the fixed backbone setting.

## Statistics

All experiments were performed in at least biological triplicate, which we define as replicates that are independent for the entirety of the experiment (i.e., from plating the cells, to treating the cells, to acquiring the data). To quantify differences in the passage 11 site frequency spectra (*Figure 4*), the Mann-Whitney test was performed using Graph Pad Prism software (nonparametric, one-tailed). This test was performed on the passage 11 site frequency spectra because this is the first passage at which we observe fixed mutations in any environment. A one-tailed test was used as we expect mutation frequency spectra to shift in a one-directional manner as mutations become fixed and the distribution becomes U-shaped. All mutation frequencies observed in the two distributions (where each distribution represents three biological triplicate data sets) to be compared were ranked. Each set of ranks was then compared to determine if the distributions are significantly different. The Mann-Whitney (*Mann and Whitney, 1947*) U values were: 2109 (HSF1 (N = 73; median = 4.92) vs. Hsp90-inhibited (N = 71; median = 3.472) non-synonymous), 661 (HSF1 (N = 41; median = 4.01) vs. Hsp90-inhibited (N = 34; median = 3.212) synonymous), 2340 (basal (N = 79; median = 4.787) vs. Hsp90-inhibited (N = 71; median = 3.472) non-synonymous), 776 (basal (N = 48; median = 3.934) vs. Hsp90-inhibited (N = 34; median = 3.212) synonymous), 2819 (HSF1 (N = 73; median = 4.92) vs. basal (N = 79; median = 4.787) non-synonymous), 970 (HSF1 (N = 41; median = 4.01) vs. basal

(N = 48; median = 3.934) synonymous). To assess statistical significance for the reverse genetics competition experiment (*Figure 6C*), an unpaired *t*-test was performed between each set of conditions, each with three biological replicate data sets (basal vs. HSF1: *p*-value =0.0019; *t* = 4.192; d*f* = 10; basal vs. Hsp90-inhibited: *p*-value =0.0091; *t* = 3.224; d*f* = 10; HSF1 vs. Hsp90-inhibited: *p*-value <0.0001; *t* = 12.45; d*f* = 10). To assess statistical significance for the CETSA (*Figure 1—figure supplement 1D*), the vehicle and STA-9090 conditions were compared across biological triplicates (technical triplicates for each biological triplicate) by an *F*-test (*p*-value <0.0001, $F(DF_n, DF_d)$=16.48 (1,194)).

## Molecular dynamics simulations

The wild type $PA_C$–$PB1_N$ complex was taken from the crystal structure with PDBID 2ZNL (*Obayashi et al., 2008*). $PA_C$ is the C-terminal PA domain (residues 257–716), and $PB1_N$ is the N-terminal domain (residues 1–15) of PB1. In 2ZNL there are some missing residues (residues 349–353; 372–397; 550–557), but they were distant from the $PA_C$–$PB1_N$ binding site. As all of these missing residues were far from the binding site they were not added back in. Furthermore, as described below, only residues 398–716 of $PA_C$ were allowed to move and the rest of $PA_C$ was position-restrained throughout the simulations. For each segment of $PA_C$, the termini were far away from the interaction site and thus were left uncapped. The N-terminus of $PB1_N$ was also left uncapped; however, the C-terminus of the short 15-residue $PB1_N$ was capped with an N-methyl group to remove the artificial negative charge. All mutations of the wild-type complex were made using the Pymol molecular modeling package (*DeLano, 2002*). All MD simulations were performed using the GROMACS 4.6.7 suite (*Hess et al., 2008*). All simulations were performed using the AMBER99 force field (*Hornak et al., 2006*) and TIP3P (*Jorgensen et al., 1983*) water. Each initial complex structure (wild-type and mutant subunits) was first immersed in a cubic box containing pre-equilibrated water molecules. The dimensions of the water box were 102 Å ×109 Å × 79 Å. Counter-ions were added as necessary to neutralize the system. The solvated system was then energy-minimized using the steepest descent algorithm for a maximum of 5000 iterations with a force constant of 1000 kJ mol$^{-1}$ nm$^{-2}$ applied to all non-hydrogen atoms. Next, a further energy minimization was performed without any constraints where the steepest descent method switched to conjugate gradient every 500 steps for a maximum of 2500 total steps. The system then underwent a two-stage equilibration process. The first stage of equilibration was a 100 ps *NVT* (isochoric–isothermal) simulation and consisted of a gentle annealing of the system from 0 to 300 K over the first 50 ps. Throughout the first stage of equilibration, a position restraint was applied to all non-hydrogen atoms with a force constant of 1000 kJ mol$^{-1}$ nm$^{-2}$. The temperature was maintained at 300 K using the V-rescale thermostat (*Bussi et al., 2007*) with a coupling time constant of 1.0 ps. The complex and solvent molecules were coupled to separate thermostats to avoid the 'hot solvent-cold solute' problem (*Cheng and Merz, 1996*; *Lingenheil et al., 2008*). The second stage of equilibration was a 500 ps *NPT* (isobaric–isothermal) simulation where position restraints were applied to all non-hydrogen atoms of residues 257–348 and 354–371 of $PA_C$, as these residues are far from the $PA_C$–$PB1_N$ interaction site. The pressure was regulated using the Berendsen barostat (*Berendsen et al., 1984*) with a reference pressure of 1 bar and a pressure coupling constant of 2.0 ps. The leapfrog algorithm (*Hockney, 1970*) with a time step of 2 fs was used for dynamics evolution. All bonds involving hydrogen were constrained using the LINCS algorithm (*Hess et al., 1997*). All neighbor searching, electrostatic interactions and van der Waals interactions were truncated at 1.4 nm. Long-range Coulomb interactions were treated using the particle mesh Ewald (PME) summation (*Essmann et al., 1995*) with a Fourier spacing of 0.12 nm and a PME order of 4. A long-range dispersion correction for energy and pressure was applied to account for the 1.4 nm cut-off of Lennard-Jones interactions (*Allen and Tildesley, 1987*). All production runs followed the same scheme as the second equilibration stage and were run for 20 ns total.

Binding free energy calculations were performed on the last 10 ns of each production simulation. The last 10 ns of each simulation were stripped of water molecules and counter ions. From these, snapshots were extracted every 10 ps giving 1000 snapshots for the last 10 ns. The binding free energy was calculated as described by Liu and Yao (*Liu and Yao, 2010*). Here, the binding free energy was calculated as $H_{total, GB} = E_{gas} + G_{sol,GB}$, and thus did not contain any contribution from solute configuration entropy. Since our focus here is on the relative order of binding affinities and the complexes have similar binding poses, the solute configuration entropy is thus omitted

(*Rastelli et al., 2010*; *Hou et al., 2011a*, *2011b*; *Wang et al., 2001*). The GBSA implicit solvent model (*Still et al., 1990*) was used with a dielectric constant of 80. The OBC(II) (*Onufriev et al., 2004*) algorithm was used to calculate the Born radii with a frequency of 1 and a cutoff of 1.4 nm. A surface tension of 3.01248 kJ mol$^{-1}$ nm$^{-2}$ was set using the Ace-approximation. Averages and standard error of mean were calculated based on 200 snapshot increments.

## Acknowledgements

We thank Brandon Ogbunugafor (University of Vermont), Jesse Bloom (Fred Hutchinson Cancer Research Center), Michael Desai (Harvard University), Kimberly Davis (MIT), and Huiming Ding (MIT) for thoughtful discussions.

## Additional information

### Funding

| Funder | Grant reference number | Author |
|---|---|---|
| National Science Foundation | Graduate Research Fellowship Program | Angela M Phillips |
| Richard and Susan Smith Family Foundation | Smith Family Foundation Excellence in Biomedical Research Award | Matthew D Shoulders |
| National Institutes of Health | NIH Director's New Innovator Award 1DP2GM119162 | Matthew D Shoulders |
| Merck | UNCF-Merck Postdoctoral Fellowship | Emmanuel E Nekongo |
| National Science Foundation | CAREER Award | Matthew D Shoulders |
| National Institutes of Health | P30-ES002109 | Matthew D Shoulders |

The funders had no role in study design, data collection and interpretation, or the decision to submit the work for publication.

### Author contributions

Angela M Phillips, Designed and performed experiments, analyzed data, drafted and edited the manuscript, and acquired funding; Luna O Gonzalez, Vincent L Butty, Stuart S Levine, Leonid A Mirny, Analyzed data, edited the manuscript; Emmanuel E Nekongo, Anna I Ponomarenko, Performed experiments, edited the manuscript; Sean M McHugh, Performed molecular dynamics simulations, edited the manuscript; Yu-Shan Lin, Supervised molecular dynamics simulations, analyzed data, and edited the manuscript; Matthew D Shoulders, Conceived the project, designed and supervised experiments and data analysis, drafted and edited the manuscript, and acquired funding

### Author ORCIDs

Angela M Phillips http://orcid.org/0000-0002-9806-7574
Yu-Shan Lin http://orcid.org/0000-0001-6460-2877
Leonid A Mirny https://orcid.org/0000-0002-0785-5410
Matthew D Shoulders http://orcid.org/0000-0002-6511-3431

### Decision letter and Author response

Decision letter https://doi.org/10.7554/eLife.28652.021
Author response https://doi.org/10.7554/eLife.28652.022

## Additional files

**Supplementary files**
• Supplementary file 1. Primer sequences for qPCR and PA sequencing.
DOI: https://doi.org/10.7554/eLife.28652.019
• Transparent reporting form
DOI: https://doi.org/10.7554/eLife.28652.020

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
