## [Decision Letter]

Thank you for submitting your article "Host Proteostasis Modulates Influenza Evolution" for consideration by *eLife*. Your article has been reviewed by three peer reviewers and the evaluation has been overseen by a Reviewing Editor and Wenhui Li as the Senior Editor. The reviewers have opted to remain anonymous.

The reviewers have discussed the reviews with one another and the Reviewing Editor has drafted this decision to help you prepare a revised submission.

Summary:

Phillips and colleagues present work demonstrating that unique cellular proteostasis environments dictate the mutational landscape of influenza A virus over serial passage in MDCK cells. The authors utilize a unique chemical/biology approach to either inhibit or activate nodes of the cellular heat shock protein family network (nuclear: HSF1 activation, cytoplasmic: Hsp90 inhibition). Influenza A/Wuhan/95 virus was chosen, and as the virus is shown to have comparable RNA dependent RNA RNA polymerase mutational kinetics to a more pathogenic H5N1 virus, studying proteostasis' role in shaping viral protein adaptation may provide insights on pathogenesis. The study reveals changes to the mutational trajectory and adaptation of influenza virus, which may be dependent on cellular proteostasis involvement

Essential revisions:

1) It is shown that viral infection itself does not activate the HSR. It is also shown that STA treatment does not activate the HSR. However, the authors need to demonstrate that viral infection + STA does not activate the HSR. It is possible that STA treatment sensitizes the cytosolic environment to viral infection, which would lead to increased HSR activation that could complicate the analysis.

2) A major question regarding this manuscript is 'Does HSF1 activation influence viral trajectories’? However, this is not clear despite statements to that effect in the discussion. It isn't clear that the HSF1 activated environment shows significant differences in the data shown in Figure 3. The stats for the HSP90 inhibited environment are shown, but no stats for the HSF1 activated environment are shown. Similarly, the data shown in Figure 5 do not show the basal environment. This makes it difficult to determine whether effects are coming from increased HSF1 activity or reduced Hsp90 activity. Even if the data are not significant between HSF1 and basal, it is still important to include the stats to help determine where these effects are coming from.

3) The method for determining viral titer is problematic. The authors use a hemagglutination assay, which measures viral particles but not infectious particles. In the event that the acquired mutations changed the ratios of infectious to un-infectious particles, the passaging would have been performed at different multiplicities of infection. This would obviously complicate the interpretation of the results. I would suggest that the authors determine plaque forming units as well as hemagglutinating units across their passages to ensure there was no change.

4) The authors are overexpressing a transcription factor (HSF1) that not only regulates the HSR pathway, but also regulates genes involved in DNA damage repair and metabolism. As such, there are expected transcriptomic changes beyond what is presented in Figure 1—figure supplement 1. The RNAseq results are extremely filtered – a global analysis needs to be shown, as other changes to the cell upon HSF1 activation may influence the response to influenza and its subsequent mutational adaptation. The same logic can be applied to the authors' Hsp90 inhibition setting, as pre-infection inhibition of a well-known nuclear chaperone likely has manifest consequences on access of transcription or nuclear factors. This setting should not be thought of as "cytoplasmic," but in fact, may have intimate connections to the authors' HSF1 setting. To clarify their system, the authors should display global transcriptomics for both chemical treatment mock (no virus) and chemical treatment infected (+ virus) settings.

---

## [Author Response]

Essential revisions:1) It is shown that viral infection itself does not activate the HSR. It is also shown that STA treatment does not activate the HSR. However, the authors need to demonstrate that viral infection + STA does not activate the HSR. It is possible that STA treatment sensitizes the cytosolic environment to viral infection, which would lead to increased HSR activation that could complicate the analysis.

This experiment was performed per the Reviewers’ request (see revised Figure 1—figure supplement 2). The resulting data show that viral infection + STA-9090 under our conditions does not activate the heat shock response.

2) A major question regarding this manuscript is 'Does HSF1 activation influence viral trajectories’? However, this is not clear despite statements to that effect in the discussion. It isn't clear that the HSF1 activated environment shows significant differences in the data shown in Figure 3. The stats for the HSP90 inhibited environment are shown, but no stats for the HSF1 activated environment are shown. Similarly, the data shown in Figure 5 do not show the basal environment. This makes it difficult to determine whether effects are coming from increased HSF1 activity or reduced Hsp90 activity. Even if the data are not significant between HSF1 and basal, it is still important to include the stats to help determine where these effects are coming from.

Data from the basal environment have been added in Figure 6 (the former Figure 5 referenced here). Additionally, statistics comparing the HSF1-activated environment to the basal environment have been added to both figures referenced.

In the case of Figure 3 (now Figure 4), although mutations do become fixed at earlier passages when HSF1 is activated, the shape of the HSF1-activated site frequency spectrum is indeed not significantly different from that of the basal environment. We added the requested statistics to Figure 4. Our discussion of this data focuses on the fact that the Hsp90-inhibited environment is different, as in the original submission.

In the case of Figure 5 (now Figure 6), all conditions are significantly different from each other and the data for the basal environment are now shown as requested. Hsp90 inhibition significantly enhances fitness of the H452Q PA variant relative to the basal environment. HSF1 activation significantly reduces the fitness of the H452Q variant relative to the basal environment. Variant fitness in the HSF1-activated and Hsp90-inhibited environments is also significantly different. This data, in addition to the data presented in Figure 5 (originally Figure 4), indicates that HSF1 activation does indeed influence viral trajectories.

3) The method for determining viral titer is problematic. The authors use a hemagglutination assay, which measures viral particles but not infectious particles. In the event that the acquired mutations changed the ratios of infectious to un-infectious particles, the passaging would have been performed at different multiplicities of infection. This would obviously complicate the interpretation of the results. I would suggest that the authors determine plaque forming units as well as hemagglutinating units across their passages to ensure there was no change.

We evaluated the infectious titer (via TCID50 assay) of each biological triplicate viral population evolved in each proteostasis environment at intermittent passages (the Editor agreed that data collected at intermittent passages was sufficient). We used this data to calculate the multiplicity of infection (MOI) throughout serial passaging, which is now shown in Figure 3—figure supplement 1. The exact infectious MOI does fluctuate slightly during serial passaging (0.001–0.04 virions/cell), as expected. However, as noted by the Reviewers, the key points are that (1) the MOI does not vary systematically between proteostasis environments and (2) the MOI is always < 0.1 virions/cell to minimize co-infection, as was intended in our experimental design. Thus, our interpretation of the results remains unchanged.

Hemagglutination titers for each passage are also still shown in Figure 3—figure supplement 1.

4) The authors are overexpressing a transcription factor (HSF1) that not only regulates the HSR pathway, but also regulates genes involved in DNA damage repair and metabolism. As such, there are expected transcriptomic changes beyond what is presented in Figure 1—figure supplement 1. The RNAseq results are extremely filtered – a global analysis needs to be shown, as other changes to the cell upon HSF1 activation may influence the response to influenza and its subsequent mutational adaptation. The same logic can be applied to the authors' Hsp90 inhibition setting, as pre-infection inhibition of a well-known nuclear chaperone likely has manifest consequences on access of transcription or nuclear factors. This setting should not be thought of as "cytoplasmic," but in fact, may have intimate connections to the authors' HSF1 setting. To clarify their system, the authors should display global transcriptomics for both chemical treatment mock (no virus) and chemical treatment infected (+ virus) settings.

We confirmed with the Editors that providing a less-filtered analysis of our current RNAseq dataset would address this point. In addition to the full RNA-Seq data set provided in Figure 2—source data 2, we now show such an analysis in our new Figure 2, which includes: (1) a heat map showing all differentially expressed transcripts (≥ 2-fold change, with p-value < 10-5) accompanied by a complete list of all transcripts that meet these thresholds in Figure 2—source data 3; (2) volcano plots showing the distribution of transcript expression for each of our treatments, labeling outliers as well as proteostasis network components, transcription factors, interferon-related genes, and DNA damage repair genes above our thresholds; (3) the original heat map that focuses on select transcripts we intend to perturb by HSF1 activation but not Hsp90 inhibition (cytosolic chaperones) and those we do not intend to perturb by either treatment (ER chaperones).

This more in-depth analysis of the transcriptomic data is now discussed in detail in the text that accompanies Figure 2, as follows:

“To further characterize these perturbed proteostasis environments, we performed RNA-Seq for each treatment in the MDCKHSF1 and MDCKYFP cell lines. […] An enhanced environment is attained by treatment with TMP, a chaperone-inhibited environment is attained by treatment with STA-9090, and a basal environment is attained by vehicle treatment.”